# MAQA: A Multimodal QA Benchmark for Negation

**Judith Yue Li**
Google Research
judithyueli@google.com

**Aren Jansen**
Google Research
arenjansen@google.com

**Qingqing Huang**
Google Research
qqhuang@google.com

**Joonseok Lee**
Google Research
& Seoul National University
joonseok@google.com

**Ravi Ganti**
Google Research
gmravi@google.com

**Dima Kuzmin**
Google Research
gmravi@google.com

## Abstract

Multimodal learning can benefit from the representation power of pretrained Large Language Models (LLMs). However, state-of-the-art transformer based LLMs often ignore negations in natural language and there is no existing benchmark to quantitatively evaluate whether multimodal transformers inherit this weakness. In this study, we present a new multimodal question answering (QA) benchmark adapted from labeled music videos in AudioSet (Gemmeke et al., 2017) with the goal of systematically evaluating if multimodal transformers can perform complex reasoning to recognize new concepts as negation of previously learned concepts. We show that with standard fine-tuning approach multimodal transformers are still incapable of correctly interpreting negation irrespective of model size. However, our experiments demonstrate that augmenting the original training task distributions with negated QA examples allow the model to reliably reason with negation. To do this, we describe a novel data generation procedure that prompts the 540B-parameter PaLM model to automatically generate negated QA examples as compositions of easily accessible video tags. The generated examples contain more natural linguistic patterns and the gains compared to template-based task augmentation approach are significant.

## 1 Introduction

Large language models (LLMs) have difficulty understanding negation in natural language. Pretrained LLMs often ignore negation in cloze questions and give same prediction for negated ("Birds cannot [MASK]") and non-negated ("Birds can [MASK]") queries (Kassner and Schütze, 2019; Hosseini et al., 2021). Hossain et al. (2022) analyzed the training corpora of state-of-the-art LLMs and found that negation is rarely present, leading to the poor handling of negation at inference time.

State-of-the-art multimodal learning leverages pretrained LLMs for fusing different modalities (Jia et al., 2021; Radford et al., 2021; Oncescu et al., 2021; Kilgour et al., 2022; Nagrani et al., 2022). Will the fine-tuned LLMs intended for multimodal applications inherit the negation problem? Huang et al. (2022) showed that the zero-shot performance on the text query based audio retrieval task degrades when the text query includes negation cues, e.g., "no vocals". Yu et al. (2022) showed that the text-to-image model generates items that are mentioned in the text prompt, even when the prompt

NeurIPS 2022 Workshop on Synthetic Data for Empowering ML Research.

suggests the absence of the item. However, there is no benchmark for quantitatively evaluation of how well negation patterns in the text are handled in such multimodal settings.

In this study, we created MAQA, a binary music audio question answering benchmark, to evaluate how well the multimodal transformers understand negation in music related questions. This benchmark is created from labeled videos in the music-related portion of AudioSet (Gemmeke et al., 2017). While the original benchmarks features 5000 hours of audios labeled with 527 audio event classes and only contains a handful of labels including negation, the proposed benchmark MAQA features a significant portion of negated questions that are generated programmatically from the original audio labels. Our goal is to evaluate if multimodal transformer can be fine-tuned to understand new concepts, e.g., "no vocals" as negation of the previously learned concept, e.g., "vocals" through compositional generalization.

The main contributions of the paper are: (1) A compositional generalization experiment that demonstrates standard fine-tuning prevents our baseline model, a multimodal transformer modified from the multilingual T5 (MT5) (Raffel et al., 2019; Xue et al., 2021) from generalizing to new concepts that are negation of learned concepts. (2) A PaLM-based data generation approach that automatically generate negated QA examples from easily access video tags. (3) Two task augmentation strategies that lead to a significant boost of the model performance on portion of MAQA with text negation.

The rest of this paper is organized as follows. Section 2 provides relevant background and related work on negation, compositional generalization and multimodal learning. Section 3 provides an overview of the MAQA dataset and its statistics. Section 4 details how we create the benchmark through data generation. The models and experiment results are presented in Section 5 and 6. The paper closes with a discussion on the limitations, implications of our results and future work.

## 2   Related Works

**Negation.** Despite improvements of LLMs in many NLP tasks such as natural language understanding, reading comprehension, zero-shot text generation, negation remains a challenge for pre-trained LLMs (Kassner and Schütze, 2019; Hosseini et al., 2021). Data augmentation has been used to tackle negation in the NLP literature. For example, modification of the natural language understanding corpora by adding negation to the verb or adjective and reversing the labels was proposed in (Hossain et al., 2020), and an unlikelihood loss for the span corruption pre-training tasks was proposed in (Hosseini et al., 2021). Negation is also addressed in the meta-learning literature (Murty et al., 2021), where it is treated as one of the reasoning categories that requires additional few-shot classification tasks to augment the original task distribution.

**Compositional Generalization.** Compositional Generalization refers to the ability to understand novel concept as compositions of previously learned concept or *atoms*. Negation can be thought as a form of composition. In the field of semantic parsing, several benchmarks have been proposed to evaluate compositional generalization (Lake and Baroni, 2018; Keysers et al., 2020; Kim and Linzen, 2020), which have encouraged development of techniques and architectures to make LLMs better at solving compositional tasks (Furrer et al., 2020; Ontanon et al., 2022; Csordás et al., 2021; Qiu et al., 2022). Several multimodal benchmarks have shown visually grounded LLMs often struggle with compositional generalization in visual reasoning tasks (Johnson et al., 2017), visual grounded command following tasks (Ruis et al., 2020), text-to-image matching (Zhang et al., 2021), etc. Our study focus on evaluating audio grounded LLMs on compositional tasks involves negation.

**Multimodal QA.** Multimodal question answering benchmarks are used to probe the multimodal models to evaluate their perception and reasoning capability on different modalities. Visual Question Answering benchmarks (Zhang et al., 2015; Agrawal et al., 2015) commonly consist of triplets of (image, a natural language question about the image, answer), and the task is to answer the question based on the visual cue in the image. In the field of audio perception, audio QA benchmarks (Fayek and Johnson, 2020) are less common than audio classification benchmarks (Gemmeke et al., 2017). In the music domain, most benchmarks are music information retrieval tasks (Law and Von Ahn, 2009), where the text labels are usually in the format of short form music tags.

**Multimodal Transformers.** A series of Transformer-based multimodal models (Sun et al., 2019; Tan and Bansal, 2019; Lu et al., 2019), referred to as "Multimodal Transformers" in this study, explored using Transformer encoder as a join encoder for multimodal fusion achieve state-of-the-art

Table 1: Examples of generated Binary Audio QA Pairs in MAQA. The original AudioSet example is a music audio clip associated with the following tags: *Bass guitar, Guitar, Acoustic guitar and Strum*. Questions and their negated counterpart are generated from the sampled attributes with the PaLM based approach. Negative attributes *steel guitar, slide guitar* are sampled from the sibling nodes in the AudioSet ontology.

| Sampled Attributes | Question | Answer | Negated |
|---|---|---|---|
| Bass guitar (+) | Q1)The musical instrument played | TRUE | No |
| Bass guitar (+) | in the song is **Bass guitar** | TRUE | No |
| Bass guitar (+) | Q2)**Bass guitar** is not played in the song | FALSE | Yes |
| steel guitar, slide guitar (−) | Q3)The song has **steel guitar** or **slide guitar** | FALSE | No |
| steel guitar, slide guitar (−) | Q4)The song does not have **slide guitar** or **steel guitar** | TRUE | Yes |

Table 2: Statistics on Music Audio QA (MAQA) Benchmark. Statistics on Music Audio QA (MAQA) Benchmark. Both evaluation sets ASBaseEval and ASNegationEval are generated by PaLM. Each training set has a template-generated and a PaLM-generated version. All the datasets have balanced binary label distributions. ASNegationEval contains ASBaseEval and its negated counterparts. The music attributes have a simialr distribution in training and evaluation split.

| Data Version | Label Stats | | # of QA Pairs | | # of mentions | | | |
|---|---|---|---|---|---|---|---|---|
| | True | Negated | Non-negated | Negated | Genre | Mood | Instrument | Role |
| ASBaseEval | 50% | 0% | 17,028 | 0 | 5574 | 730 | 9740 | 984 |
| ASNegationEval | 50% | 50% | 17,028 | 17,028 | (32.7%) | (4.3%) | (57.2%) | (5.8%) |
| ASBaseTrain | 50% | 0% | 1,263,004 | 0 | 439,904 | 28,634 | 740,126 | 54,340 |
| ASNegationTrain | 50% | 50% | 1,263,004 | 1,263,004 | (34.8%) | (2.3%) | (58.6%) | (4.3%) |

results on a range of multimodal QA tasks. Changpinyo et al. (2022) proposed a multimodal version of T5 (Raffel et al., 2019). Given the image and the question in a VQA example, the multimodal T5 takes the global and regional image features generated by a pre-trained visual encoder and text tokens of the question as inputs, and solves a classification problem with pre-defined classes of answers for the VQA task. The parameters of the visual encoder are frozen during T5 fine-tuning. We follow the same approach but use pre-trained audio encoders (Gemmeke et al., 2017; Huang et al., 2022) to extract global representation of the music audio. A detailed survey of audio representation can be found in Huang et al. (2022).

## 3   Music Audio Question Answering (MAQA)

To evaluate the ability of multimodal models to reason with negation, we create a music audio QA benchmark (MAQA) which emphasizes on correct understanding of text negation. The music audio QA pairs are generated programmatically from the music related portion of AudioSet (Gemmeke et al., 2017), which contains music audio clips annotated with music attributes and an ontology describing their relationship. There are $388, 262$ and $4, 497$ unique music audio clips in the train and evaluation split, respectively. Each clip is labeled with one or more music tags out of the $141$ unique music attributes covering music genres, music roles, music instruments and music moods.

Table 1 presents an example in MAQA, which consists of four QA pairs generated from an example of music attribute labeled audio clip in AudioSet. Q1 and Q2 are questions generated from the same seed attribute, and essentially probe about the same musical skill, i.e., listen to a music audio and try to identify if a bass guitar is played. Q2 is a negated form of Q1. If a model answer Q1 correctly and fail on its negated counterpart Q2, it suggests that the model does not understand the negation logic in the question and unable to perform compositional generalization.

MAQA contains two evaluation sets ASBaseEval and ASNegationEval and two training sets ASBaseTrain and ASNegationTrain as shown in Table 2 with balanced binary label distribution, featuring QA pairs about music moods, genre, instrument and roles. ASBaseTrain / ASBaseEval contains non-negated QA pairs about music audio recordings. ASNegationTrain / ASNegationEval is a superset of ASBaseTrain / ASBaseEval, and it also includes their negated counterparts of the QA pairs. A multimodal model with strong music audio understanding capabilities should score high on

ASBaseEval. Moreover, to demonstrate its ability of reasoning about negation logic, it has to also score high on ASNegationEval.

## 4  Data Generation

Since music descriptive text that involves negation rarely occur in the standard text corpora Hossain et al. (2022), we propose the following 3-step approach to programmatically generate binary audio QA pairs that involve text negation: 1. For each music audio-attribute pair in the original dataset, we sample a negative attribute that is not associated with the audio clip. 2. Convert the positive and the negative audio-attribute pair into a binary AQA example in the format of a triplet (audio clip, question on the attribute, *True / False* label). 3. Perform a text negation on the question and flip the binary label simultaneously to create *negated* audio QA pairs. As a first attempt we curate MAQA from AudioSet with this method, however it can be applied to other datasets containing annotated music audios. Next, we discuss the details of how we followed the 3 steps to create MAQA from AudioSet.

**Negative Attribute Sampling.**  We adopt negative sampling to create a balanced binary label distribution. In particular, we sample hard negative attributes using sibling nodes in the ontology tree and assign *False* label to the derived audio QA pair. Consider the example in Table 1, the audio clip is tagged with *Bass guitar* and *Acoustic guitar*, which are both under the parent node *Guitar*. We sample hard negative attributes *steel guitar* and *slide guitar* from the sibling nodes, to create a negative audio-attribute pair. This hard negative sampling approach encourages the model to differentiate related but different music concepts.

**Question Generation.**  We explore the following two approaches to generate questions from the audio-attribute pair sampled from the first step. The first approach is template based, and it takes advantage of the AudioSet ontology, where each music attribute is associated with one of the four attribute types: genres, roles, instruments, and moods. We use type-specific templates to convert attributes into a true-or-false question, e.g., "The *<Attribute Type>* of the song is *<Attribute Value>*.". The second approach leverages the few-shot text generation capability of PaLM (Chowdhery et al., 2022) to improve the diversity of generated questions. Similar to GPT-3 (Brown et al., 2020), when prompted with an instruction, e.g., "Generate a sentence about music given the music attribute", PaLM learns from a few demonstrations and generates questions on unseen attributes.

**Task Augmentation with Negation.**  The template-based approach convert a question to the negation form by inserting a modifier *not* before the noun, i.e., "The <Attribute Type> of the song is *not* <Attribute Value>." and the binary label is flipped. One of the limitation of this approach is that it is attribute type specific and only modifies nouns. PaLM based method overcomes the limitation as with few shot learning the model can generate different negation patterns by modifying both nouns and verbs. For example, the negation patterns associated with the instrument attribute "guitar" include "no guitar", "guitar is not played", and "the song does not feature bass guitar". For each music attribute, we use PaLM to generate a few question candidates and manually pick the best one. Row 2 and 4 in Table 1 are example questions generated in this way. More example questions generated by PaLM and the prompts used are shown in Appendix 8.3.

## 5  Multimodal Modeling

Following the VQA literature (Changpinyo et al., 2022; Zhang et al., 2015), we treat the audio QA as a binary classification task. We adopt a multimodal T5 architecture similar to that in (Changpinyo et al., 2022) to fuse the audio and text inputs, and we replace T5 with its multilingual version MT5. Each music audio clip input is represented as a 128-dimensional embedding obtained either from VGGish (Gemmeke et al., 2017)[1], which uses a VGG ConvNet architecture, or the transformer based MuLan model (Huang et al., 2022). The audio encoders are frozen when we finetune the multimodal T5. The audio embeddings are projected to the text token embedding space through a linear projection layer, which is initialized randomly at the beginning of finetuning. Then, the audio token and text token are fed into the pre-trained multi-layer MT5 (Xue et al., 2021) encoder as a sequence of vectors

---

[1] https://github.com/tensorflow/models/tree/master/research/audioset/vggish

and the final multimodal representation is classified into the binary classes. The multimodal code is based on the Flaxformer framework[2]. Training details can be found in Appendix 8.1.

# 6 Experiments and Results

We report experiment results on the ASBaseEval and ASNegationEval evaluation benchmark in Table 3 and Table 4 respectively. The Audio QA task is formulated as a binary classification problem, and we report the best AUC-ROC score and the corresponding accuracy in the positive class. To evaluate model's ability to generalize compositionally so that it can understand composed music concepts like "no vocals" that involve negation, we split the data into train and test sets following the design recommended by (Keysers et al., 2020). By design the music attributes or *atoms* are similarly represented in the train and test sets, while the test set contains novel combinations of the *atoms* that are not seen in the train set. *Compound Divergence* (CD) is used to measure quantitatively how different is the compound distributions in the train and test split (Keysers et al., 2020), while in our case CD is used as a qualitative measure (Tabel 5 in Appendix 8.2), and *compound* refers to the QA pairs after applying compositional rules, e.g., negation to the *atoms*. For each split scenario, we compare the performance of finetuned multimodal transformer with different audio feature extractors, as well as with different sized pre-trained MT5 model. Furthermore, we vary the types of QA pairs (template-based or PaLM-based) used in training split and study how compound divergence affects learning negation.

## 6.1 Music Understanding

Table 3(a) shows the result for the first split scenario where the model is trained and evaluated on non-negated QA pairs generated by PaLM. This Low CD experiment establish a fine-tuning baseline on basic music concepts. The fine-tuned multimodal MT5 score over $90\%$ AUC-ROC on the ASBaseEval benchmark that features Audio QA tasks on music styles, moods, genres, instruments, etc. Recall the random baseline is $50\%$ for balanced binary classification tasks, this suggests multimodal transformer learn to efficiently fuse audio and text signals through fine-tuning, even it is warm started from a text-only checkpoint. Probing the model on different music attributes suggests that music concepts like "Scary music", "Children music" and popular percussion instruments like "Cowbell" are easy for the fine-tuned model to pick up, while the model has a harder time to understand electronic music genres such as "Drum and bass", "Trance music".

We further replace the training examples generated by PaLM with the template-generated QA examples resulting in the Medium CD setting. The model scores around $6\%$ lower in the Medium CD setting (Table 3(b)) compared to the Low CD setting. This suggest the model can still transfer most of the music knowledge learned in a different linguistic context via compositional generalization. For both split scenarios the best multimodal model is the MT5-XL with Mulan embedding as audio features.

## 6.2 Reasoning with Negation

For the third split scenario (Table 3(b)) we apply the same fine-tuning setup as in Table 3(a) but evaluate on ASNegationEval, where the non-negated half is from ASBaseEval and the other half contains their negation counterparts. As shown in Table 4(a), the multimodal MT5 fine-tuned on non-negated audio QA pairs (ASBaseTrain) scores only $50\%$ on the ASNegationEval benchmark in this High CD setting. Although the model still scores around $80\%$ on the non-negated QAs (comparable to the accuracy on ASBaseEval in Table 3), it scores only around $20\%$ on their negated counterparts. The model does worse than the $50\%$ random guess baseline on these negated questions after fine-tuning. This shows that while the model is trained to answer the non-negated questions correctly, they also learn to "ignore" the negation cue in the negated questions. We also show that increasing the model size does not improve the AUC-ROC score, suggesting that even larger model fail to generalize compositionally using the standard fine-tuning approach.

---

[2]https://github.com/google/flaxformer

Table 3: Accuracy on ASBaseEval for two different Compound Divergence (CD) settings.

| | | Finetuning Details | | ASBaseEval | | |
|---|---|---|---|---|---|---|
| | Model | Train Data - QA Type | CD Type | AUC | Acc | |
| a) | MT5-Base+VGGish | ASBaseTrain-PaLM | Low CD | 0.905 | 0.821 | |
| | MT5-XL+VGGish | ASBaseTrain-PaLM | Low CD | 0.911 | 0.827 | |
| | MT5-Base+MuLan | ASBaseTrain-PaLM | Low CD | 0.913 | 0.828 | |
| | MT5-XL+MuLan | ASBaseTrain-PaLM | Low CD | **0.918** | 0.832 | |
| b) | MT5-Base+VGGish | ASBaseTrain-Temp | Med CD | 0.847 | 0.771 | |
| | MT5-XL+VGGish | ASBaseTrain-Temp | Med CD | 0.850 | 0.765 | |
| | MT5-Base+MuLan | ASBaseTrain-Temp | Med CD | 0.845 | 0.766 | |
| | MT5-XL+MuLan | ASBaseTrain-Temp | Med CD | **0.851** | 0.767 | |

Table 4: Accuracy on ASNegationEval for three different Compound Divergence (CD) settings.

| | | Finetuning Details | | ASNegationEval | | | |
|---|---|---|---|---|---|---|---|
| | | | | | | Acc | |
| | Model | Train Data - QA Type | CD Type | AUC | Avg | Neg | NoNeg |
| a) | MT5-Base+VGGish | ASBaseTrain-PaLM | High CD | 0.524 | 0.513 | 0.218 | 0.803 |
| | MT5-XL+VGGish | ASBaseTrain-PaLM | High CD | 0.525 | 0.525 | 0.247 | 0.802 |
| | MT5-Base+MuLan | ASBaseTrain-PaLM | High CD | **0.553** | 0.541 | 0.273 | 0.802 |
| | MT5-XL+MuLan | ASBaseTrain-PaLM | High CD | 0.528 | 0.520 | 0.220 | 0.819 |
| b) | MT5-Base+VGGish | ASNegationTrain-PaLM | Low CD | 0.896 | 0.814 | 0.814 | 0.813 |
| | MT5-XL+VGGish | ASNegationTrain-PaLM | Low CD | 0.903 | 0.821 | 0.821 | 0.821 |
| | MT5-Base+MuLan | ASNegationTrain-PaLM | Low CD | 0.905 | 0.821 | 0.821 | 0.822 |
| | MT5-XL+MuLan | ASNegationTrain-PaLM | Low CD | **0.907** | 0.825 | 0.825 | 0.825 |
| c) | MT5-Base+VGGish | ASNegationTrain-Temp | Med CD | 0.784 | 0.715 | 0.690 | 0.741 |
| | MT5-XL+VGGish | ASNegationTrain-Temp | Med CD | 0.823 | 0.743 | 0.724 | 0.763 |
| | MT5-Base+MuLan | ASNegationTrain-Temp | Med CD | 0.805 | 0.739 | 0.723 | 0.755 |
| | MT5-XL+MuLan | ASNegationTrain-Temp | Med CD | **0.828** | 0.750 | 0.740 | 0.759 |

## 6.3 Task Augmentation

We then apply task augmentation during training by augmenting ASBaseTrain with negated QA example generated by PaLM (AsNegationTrain-PaLM), which lower the compound divergence. The task augmentation proves to be an effective strategy for tackling negation. As shown in Table 4(b), multimodal MT5 fined-tuned with task augmentation improves the baseline on ASNegationEval as shown in Table 4(a) by nearly $40\%$, while obtaining similar performance on the non-negated QA pairs (ASBaseEval). The AUC-ROC score and accuracy is on par with the scores on the non-negated Audio QA pairs (ASBaseEval), suggesting that task augmentation can indeed help the model to learn to answer the questions with negation correctly. The best result on ASNegationEval is obtained by fine-tuned MT5-XL with MuLan audio embedding.

## 6.4 Template versus PaLM

We further explore how different task augmentation strategy affects the learning outcome. As shown in Table 4(c), we use template-based approach for composing QA pairs and task augmentation, and compare with the fine-tuning results with PaLM-generated QA pairs. The Template-based fine-tuning scores around $10\%$ lower in AUC score compared to PaLM-based fine-tuning. The observed gap can be explained by the increased compound divergence between the training data and the evaluation data. The accuracy difference on the non-negated split is around $7\%$ while the difference on the negated split is around $10\%$ to $12\%$. Recall that the template-based approach only modifies the noun for negation while Palm-based approach incorporates more variations, which can explain why the template-based fine-tuning performs worse on the negated split. This experiment has highlighted the importance of composing augmented tasks with natural linguistic variations that match human language used in production environment. However, even Template-based task augmentation can improve negation understanding significantly, on average $30\%$ higher than training without task augmentation (Table 4(a)).

# 7 Conclusion

In this work, we propose a new Binary Audio QA benchmark MAQA in the music domain to probe the state-of-the-art multimodal models on understanding negation. MAQA fills in the gap of lacking negation-focused evaluation benchmark in the multimodal setting. Our experiments show that standard fine-tuning prevents the multimodal transformers from generalizing to new concepts that are negation of the learned concepts. While increasing the model size or adopting a better audio encoder doesn't help with negation, task augmentation allows the model to reason with negation by providing more fine-tuning examples that contain negation. And LLMs like PaLM can be used to generate negated examples with more natural linguistic variations, which have a significant effect on the learning outcome. With the MAQA benchmark, we hope to encourage multimodal research community to develop new modeling frameworks or algorithms to handle complex natural language instructions that involves negation. We plan to release the MAQA dataset on Github.

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

# 8 Appendix

## 8.1 Training Details

For the MT5-base encoder there are 12 Transformer encoder layers and the input embedding dimension is 768. For MT5-XL there are 24 Transformer encoder layers and the input embedding dimension is 2048. MT5-XL has 3.7 billion parameters and MT5-base has 580 million parameters. The batch size is 64 for MT5-base model and 128 for MT5-XL model. We use fixed a learning rate among $\{10^{-3}, 10^{-4}, 5^{-5}\}$ and observe $1 \times 10^{-3}$ works best in general. The model outputs 2-dimensional logits as the Audio QA task is formulated as binary classification. We train all models with data parallelism using 16 Cloud TPU Pods. For all experiments we run for $50,000$ steps and reports the AUCROC and Accuracy based on the best checkpoint measured by AUCROC. For each experiment we pick the highest AUCROC of multiple runs. It takes around 1 hour for MT5-Base and around 4 hours for MT5-XL.

## 8.2 Compound Divergence

As PaLM-generated composition does not depend on rules to combine different atoms, hence it's hard to compute CD directly. Here we use Compound Divergence as a qualitative instead of quantitative measure of the difference of composition in the train and test split as shown in Table 5.

| Attribute | Train Examples | Test Examples | CD Type |
|---|---|---|---|
| **(a) Without Task Augmentation** | | | |
| Banjo | The song is played with a **banjo**. (ASBaseTrain-PaLM) | The song is played with a banjo. (ASBaseEval-PaLM) | Low CD |
| Blues | The [genre] of the song is [**blues**]. (ASBaseTrain-Temp) | The song is a blues song. (ASBaseEval-PaLM) | Med CD |
| Scary | The music is **scary**.

(ASBaseTrain-PaLM) | The music is **scary**.
The music is not **scary**.
(ASNegEval-PaLM) | High CD |
| **(b) With Task Augmentation** (ASNegationTrain) | | | |
| Chant | The music is a **Chant**.
The music is not a **Chant**.
(ASNegTrain-PaLM) | The music is a **Chant**.
The music is not a **Chant**.
(ASBaseTrain-PaLM) | Low CD |
| Dance music | The [music role] of the song is [**Dance music**].
The [music role] of the song is not [**Dance music**].
(ASNegTrain-Temp) | The music is suitable for **dancing**.

The music is not suitable for **dancing**.

(ASBaseTrain-PaLM) | Med CD |

Table 5: The train and test split design of MAQA. For each of the 5 split scenarios we list a few example questions included in the train and test split. All the QA examples or *compounds* are composed from the seed music attributes or *atoms* via Template-based (Temp) or PaLM-based (PaLM) approach. Compound Divergence (CD) Type is used to measure the difference between the train and test compound distribution. Task Augmentation is applied during training for the ASNegationEval-LowCD and ASNegationEval-MedCD split scenario.

## 8.3   Text Prompts used for PaLM

After a few iterations, we've found the text prompts shown in Table 6 works best for generating negated and non-negated QA pairs. The one-line instruction at the beginning of the prompt is crucial to guide PaLM to generate desired sentences. Table 7 shows more PaLM-generated questions. The total number of seed attributes used for text generation are 130, and for each attribute we pick one non-negated and one negated question selected from the candidate questions generated from PaLM.

| **User Input** (**Music Tag**): *Trumpet* |
| --- |

**Model Input:**

Generate a sentence about music and its negation given the music tag
music tag: drum
describe the music: The music is played with a drum
negate the description: drum is not played in the song
music tag: rock
describe the music: This is a rock song
negate the description: the genre for the song is not rock
music tag: happy
describe the music: The music makes one feel happy
negate the description: The mood of the song is not happy
music tag: Choir
describe the music: The song features a choir
negate the description: The song does not feature a choir
music tag: Tabla
describe the music: Tabla is played in the song
negate the description: Tabla is not played in the song
music tag: *Trumpet*
describe the music: *<extra_id_1>*
negate the description: *<extra_id_2>*

**Model Output:**

*<extra_id_1>*: A trumpet is played in this song
*<extra_id_2>*: A trumpet is not played in this song

Table 6: Text prompts used for PaLM to generate non-negated and negated sentences from a seed music attribute. Here we show a example output for the attribute *Trumpet*.

| Seed Attribute | Generated Questions | Generated Negated Questions |
|---|---|---|
| A capella | The song is sung without any musical instruments | The song is not sung without any musical instruments |
| Accordion | The musical instrument of the song is accordion | accordion is not played in the song |
| Acoustic guitar | The song is played with an acoustic guitar | The song is not played with an acoustic guitar |
| Ambient music | The song is ambient music | The song is not ambient music |
| Angry music | The song makes one feel angry | The mood of the song is not angry |
| Background music | The song is background music | The song is not background music |
| Bagpipes | The musical instrument of the song is Bagpipes | Bagpipes is not played in the song |
| Banjo | The song is played with a banjo | The song is not played with a banjo |
| Bass drum | The musical instrument of the song is Bass drum | Bass drum is not played in the song |
| Bass guitar | The musical instrument of the song is Bass guitar | Bass guitar is not played in the song |
| Beatboxing | Beatboxing features in the song | Beatboxing does not feature in the song |
| Bell | The musical instrument of the song is bell | bell is not played in the song |
| Bluegrass | The genre of the song is bluegrass | bluegrass is not the genre of the song |
| Blues | The song is a blues song | the song is not a blues song |
| Bowed string instrument | The musical instrument of the song is a bowed string instrument | The music is not played with a bowed string instrument |
| Brass instrument | The musical instrument of the song is brass | brass is not played in the song |
| Carnatic music | The music is Carnatic | The music is not Carnatic |
| Cello | The musical instrument of the song is Cello | Cello is not played in the song |
| Chant | The music is a chant | The music is not a chant |
| Choir | The song features a choir | The song does not feature a choir |
| Christian music | It is Christian music | It is not Christian music |
| Christmas music | The music is Christmas music | The music is not Christmas music |
| Clarinet | The song is played by clarinet | clarinet is not played in the song |
| Classical music | The music is classical | The genre for the song is not classical |
| Country | The genre of the song is Country | This is not a Country song |
| Cowbell | The song has cowbell | The song does not have cowbell |
| Cymbal | The musical instrument of the song is Cymbal | Cymbal is not played in the song |
| Dance music | The music is suitable for dancing | The music is not suitable for dancing |
| Didgeridoo | The music is played on a didgeridoo | The music is not played on a didgeridoo |
| Disco | The song belongs to the disco genre | The song does not belong to the disco genre |
| Double bass | The musical instrument of the song is double bass | double bass is not played in the song |
| Drum and bass | The music is drum and bass | The song is not drum and bass |

Table 7: More examples of PaLM-generated non-negated and negated questions.

