# OpenReview forum: "MAQA: A Multimodal QA Benchmark for Negation"
_NeurIPS.cc/2022/Workshop/SyntheticData4ML — Neurips 2022 SyntheticData4ML_

### Official Review · Reviewer_Ltqp · 2022-10-17
**The authors extend AudioSet with negations and will release the dataset.**

**Rating:** 5
**Confidence:** 3

**Review:**

The authors proopose a benchmark of labeled music videos (adapted from an existing one - AudioSet) that evaluates whether a model can learn new concepts as negations of already learned concepts. The authors automatically augment AudioSet with negated QA pairs. The contribution is more as a resource (extended AudioSet with negation) than theoretical.

It would be good to briefly introduce AudioSet in the introduction (e.g. around last paragraph of page 1). This will help with understanding the experiments and results better.

Minor comment - it would be better if Table 1 and 2 are on page 3 (where they are first mentioned) instead of page 4.

How are Low-CD, Med-CD. High-CD quantified?

In Table 4, some of the accuracies are not bolded correctly. What is the state of the art?

What other benchmarks can be created with your method? Can it generalize beyond negation?

The paper has some grammatical errors but it somewhat lacks clarity. The main contribution is the new dataset and I would expect some more metrics and statistics, evaluation of more models (rather than just MT5), etc.

---

### Official Review · Reviewer_4mWT · 2022-10-17
**Interesting direction, but improvements needed**

**Rating:** 3
**Confidence:** 4

**Review:**

This paper introduces a new benchmark on a multi-modal binary classification task on audio clips. Specifically, the authors propose a task which takes a pair of audio clip and a natural-language statement as an input and aims to predict if the statement is correct or not. Given the nature of this binary classification problem, this work focuses on the negation problem which was introduced in the synthetic data by templates or prompting with a large language model. The experimental results show that 1) more training data with negation helped to improve the performances; and 2) the prompting-based method was better than template-based approach.

Both high-level idea on the problem formulation and detailed methods of data synthesis and model development sound reasonable. But I have the following major concerns that would need to be corrected or improved:
* While the authors claim that this is a yes/no QA problem, the actual data include no questions, but statements. I think it can be corrected by 1) generating actual questions instead of statements; or 2) referring this problem as a multi-modal inference or entailment problem instead.
* Since both training and evaluation datasets are synthesized with the same methods, the performance improvements by adding the negation cases are expected and not surprising. To make the results more interesting and meaningful, I suggest to add some more realistic test data from any existing data or crowdsourcing in addition or instead of the synthetic one.
* More qualitative analysis on the generated statements would help readers to understand better about the challenges of this task. In addition, some error analysis from the best model results would tell more about some room for improvements in the future.

---

### Official Review · Reviewer_6yeu · 2022-10-19
**Data augmentation to train multimodal QA model to generalize better to negation in queries**

**Rating:** 8
**Confidence:** 4

**Review:**

Authors aim to improve the ability of multimodal retrieval systems to handle negative attributes ("music not containing a base guitar"). They find that standard training generalizes poorly to negative attributes, presumably because they are rarely used in training texts.  They then propose to augment the training data with negative attributes using either a template-based approach or an large LM with query engineering to elicit negative queries.  The negative attributes are inferred from the regular (positive) attributes contained in the metadata of an audio data set.  They find that this dramatically improves the generalization to negative-attribute queries, with the LM-paraphrasing method beating the template-based approach.  The results are pretty much as expected, given that the LM method generates more variation in the synthetic training data.

The paper is well-written, the methodology careful and thorough.

---

### Meta-Review · Area_Chair_6dzC · 2022-10-20

**Recommendation:** Accept

**Review:**

There is reasonable variance in reviewer scores, but I tend to agree that this paper is well written and contains a sensible and promising approach to highlighting and improving issues with semantic understanding of negation in these models. I would suggest the authors take on board the feedback of all the reviewers and consider including some additional experiments and in particular qualitative analysis to help readers better understand results.